# Isolation and Characterization of Phosphate Solubilizing *Streptomyces* sp. Endemic from Sugar Beet Fields of the Beni-Mellal Region in Morocco

**DOI:** 10.3390/microorganisms9050914

**Published:** 2021-04-24

**Authors:** Yassine Aallam, Driss Dhiba, Sanaâ Lemriss, Amal Souiri, Fatma Karray, Taoufik El Rasafi, Nezha Saïdi, Abdelmajid Haddioui, Saâd El Kabbaj, Marie Joëlle Virolle, Hanane Hamdali

**Affiliations:** 1Laboratory of Biotechnology and Valorization of Plant Genetic Resources, Faculty of Sciences and Technology, University of Sultan Moulay Slimane, P.O. 523, Beni-Mellal 23000, Morocco; yassine.aallam@gmail.com (Y.A.); elrasafi_taoufik@hotmail.com (T.E.R.); ahaddioui@yahoo.fr (A.H.); 2International Water Research Institute, University Mohammed 6 Polytechnic (UM6P), Moulay Rachid, Ben Guerir 43150, Morocco; driss.dhiba@um6p.ma; 3Laboratory of Research and Medical Analysis of Gendarmerie Royale, Department of Biosafety PCL3, Rabat, Morocco; slemriss@lram-fgr.ma (S.L.); asouiri@lram-fgr.ma (A.S.); selkabbaj@lram-fgr.ma (S.E.K.); 4Center of Biotechnology of Sfax (CBS), Laboratory of Environmental Bioprocesses (LBPE), Sfax BP: 1177-3018, Tunisia; karray.fatma@gmail.com; 5HTMR Laboratory, University Mohammed 6 Polytechnic (UM6P), Moulay Rachid, Ben Guerir 43150, Morocco; 6CRRA Rabat, Plant Breeding, Conservation and Valorization of Plant Genetic Resources Research Unit, B.P: 6356-Rabat Institutes, Rabat 415, Morocco; nezsaidi@yahoo.fr; 7Institute for Integrative Biology of the Cell (I2BC), Université Paris-Saclay, CEA, CNRS, 91198 Gif-sur-Yvette, France

**Keywords:** insoluble phosphate, biosolubilizing, biofertilizer, Actinobacteria, sugar beet

## Abstract

In the course of our research, aimed at improving sugar beets phosphorus nutrition, we isolated and characterized *Streptomyces* sp. strains, endemic from sugar beet fields of the Beni-Mellal region, which are able to use natural rock phosphate (RP) and tricalcium phosphate (TCP) as sole phosphate sources. Ten *Streptomyces* sp. isolates yielded a comparable biomass in the presence of these two insoluble phosphate sources, indicating that they were able to extract similar amount of phosphorus (P) from the latter for their own growth. Interestingly, five strains released soluble P in large excess from TCP in their culture broth whereas only two strains, BP, related to *Streptomyces bellus* and BYC, related to *Streptomyces enissocaesilis*, released a higher or similar amount of soluble P from RP than from TCP, respectively. This indicated that the rate of P released from these insoluble phosphate sources exceeded its consumption rate for bacterial growth and that most strains solubilized TCP more efficiently than RP. Preliminary results suggested that the solubilization process of BYC, the most efficient RP and TCP solubilizing strain, involves both acidification of the medium and excretion of siderophores. Actinomycete strains possessing such interesting RP solubilizing abilities may constitute a novel kind of fertilizers beneficial for plant nutrition and more environmentally friendly than chemical fertilizers in current use.

## 1. Introduction

Phosphorus (P) is one of the 16 elements essential for plant growth [1]. Phosphate availability greatly determines growth and fitness of the plants and thus crops quality and yields [2]. In condition of P deficiency, plant root development is inhibited and this leads to a delay in plant growth [3]. Phosphorus concentration in natural soils varies from 50 to 3000 mg kg^−1^ of soil, yet only 0.1% of total phosphorus is really accessible to plants [4,5], Indeed, most phosphate is immobilized in the soil [6] either via its adsorption on soil particles, precipitation with various minerals (Al, Fe and Ca) or interconversion into organic forms by soil-born micro-organisms [7]. To overcome this problem, nearly two millions tons of soluble chemical phosphate fertilizers are spread each year on agricultural fields worldwide [8]. However, still a significant fraction of these fertilizers is converted into insoluble forms [9].

The increasing awareness of environmental issues linked to agrochemical inputs stimulates the development of a sustainable agriculture and the replacement or complementation of the chemical fertilizers by other more ecological and environmentally friendly processes [10]. The direct use of natural rock phosphate in traditional agriculture in Morocco is one of such processes [11]. Morocco contains three quarters of the world’s rock phosphate (RP) reserves [12]. RP is an hydroxyapatite (Ca_10_(PO_4_)_6_ CaF_2_) not directly usable by plants [13] except in some acidic soils or in soils rich in specific micro-organisms able to convert insoluble phosphate into a form easily assimilable by plants [14]. Several reports demonstrated that these P-solubilizing microorganisms (PSM) could increase growth and crops yields of several agricultural plants including wheat [15], chickpea [16], rice and tomato [17] and can potentially be used as P-biofertilizers.

Sugar beet(*Beta vulgaris* L.) is the main industrial crop grown in the vast agricultural lands of the Beni-Mellal region of Morocco and constitutes 21.2% of the national production of sugar beet [18]. Since increasing growth and yield of this economically important crop is a constant concern, we investigated the presence of PSM endemic to these specific soils.

Among these PSM, Actinobacteria, including Actinoplanes, Streptomyces and *Micromonospora* [7,13], are of special interest since besides their PSM abilities, they also produce bioactive secondary metabolites able to limit growth of various phytopathogens agents [9,15,19,20,21,22] or molecules stimulating growth or eliciting natural plant defenses [23,24]. Exploring the richness of endemic PSM Actinobacteria in the soils specifically used for sugar beet cultivation is of interest to develop adequate new bio-fertilizer agents to stimulate sugar beet growth in the Beni-Mellal region. We thus screened for and isolated Actinomycetes endemic to these sugar beet rhizospheric soils that were able to grow and release soluble phosphate from insoluble phosphate sources in laboratory conditions. Putative solubilization mechanisms used by these bacteria were discussed and taxonomic characterization of the most efficient solubilizing isolates was achieved.

## 2. Materials and Methods

### 2.1. Location and Collection of Soil Samples

The area of interest is located in the Beni-Mellal region of Morocco (3220′22″ N, 6°21′39″ W), that has an altitude of approximately 400 m. This region is located in the irrigated perimeter of Beni Amir-Beni Moussa separated by the Oum er Rbia river [25] that creates two independent irrigated perimeters: Beni-Moussa and Beni-Amir (Figure 1; Table 1) of 33,000 ha and 69,500 ha, respectively [26]. This region is characterized by a semi-arid climate with an average rainfall generally below 280 mm [27] and an average temperature of 19 °C [28].

Soil samples were collected in June 2017 from three different fields (Figure 1; Table 1). From each sampling point, 3 cm surface residues were first removed and three subsamples distant of 10 m from each other and in different directions were collected from 0 to 10 cm depth and thoroughly mixed to ensure samples homogeneity. All soil samples were then air dried, homogenized, sieved (<2 mm), placed in a sterile tightly closed polyethylene bag, stored at 4 °C and the soil was used for further experiments within 48 h.

### 2.2. Chemical Analysis

Conductivity and pH (in water) were measured in a soil-water suspension (1: 5 *v*/*v*). Potential acidity was determined after dilution of the soil sample in a suspension of potassium chloride KCl 1N [29]. The organic (OM) and mineral (MM) matter were determined by ignition at 550 °C for 5 h in a furnace, as described in [27]. Soil moisture was assessed by measuring the evolution of the mass of 10 g of each sample maintained in an oven at 105 °C for 24 h [29]. The Kjeldahl method was used to determinate the total concentration of nitrogen. The determination of the phosphorus (P_2_O_5_) content was carried out by ICP-AES (Inductively coupled Plasma Atomic Emission Spectrometry, Perkin Elmer Wellesley, Waltham, MA, USA) [30].

### 2.3. Isolation of Total Flora and of Actinomycetes

Two grams (wet weight) of each soil sample were suspended in 18 mL of sterile physiological serum (9 g/L, NaCl), homogenized and sonicated as described previously [31]. Next, 0.1 mL of various dilutions of the treated samples was plated in triplicate on the surface of nutrient agar (Difco, Sparks, MD, USA) for Gram positive and Gram negative bacteria and of synthetic minimum medium (SMM) containing 10 g/L glucose, 2 g/L NaNO3, 0.5 g/L MgSO4.7H2O, 0.5 g/L KCl, 0.01 g/L FeSO4.7H2O and K2HPO4 (0.5 g/L, 4.38 mM) as described previously [15]. The pH of SMM was adjusted to 7 and it was sterilized at 121 °C for 20 min. This medium was supplemented with 40 mg mL/L actidione and 10 mg/mL nalidixic acid to inhibit growth of fungi and Gram negative bacteria, respectively. After plating, the agar plates were incubated for 21 days at 28 °C in order to allow growth of the slow growing Actinomycetes. Actinomycetes were recognized on the basis of their morphological features described in the International Streptomyces Project (ISP) [32].

### 2.4. Screening for Actinomycetes Able to Use Rock Phosphate (RP) and Tricalcium Phosphate (TCP) as Sole Phosphorus Source

Selection of Actinomycetes able to use RP originating from Khouribga phosphate mine in Morocco [15], as sole P source, was carried out by plating 300 colonies (100 colonies from each investigated soil) on the SMM containing 0.5 g/L of RP (approximately equivalent to 2.2 mM phosphorus) as a unique P source or on the SMM containing soluble K_2_HPO_4_ (0.5 g/L, 4.38 mM) or no P source. Spores of Actinomycete isolates showing the most active growth on SMM containing RP as sole P source were stored in 20% (*w*/*v*) sterile glycerol at −20 °C and were subsequently tested for their ability to grow on SMM containing TCP (0.5 g/L, Ca_3_(PO_4_)_2_) (Sigma Aldrich, Isère, France) as the sole P source.

### 2.5. Quantitative Estimation of the Amount Soluble Phosphate Released in the Growth Medium by the Selected Actinomycete Strains

The selected Actinomycete isolates were inoculated at 10^6^ spores/mL in 250 mL Erlenmeyer flasks containing 50 mL of liquid SMM medium with 0.5 g/L RP or 0.5 g/L TCP as sole P source, in triplicate, and grown for 5 days at 28 °C on a rotary shaker (180 g/min) [15]. Every day, a 1 mL aliquot of each culture was taken and centrifuged at 10,000× *g* for 10 min, dry biomass as well as the pH of the supernatant were determined. The supernatant was analyzed for P_2_O_5_ content by the chlorostannous reduced molybdo-phosphoric acid blue color method [33]. Similar measures were carried out in non-inoculated flasks incubated in the same conditions to determine the amount of phosphate spontaneously released from RP and TCP.

### 2.6. Production and Detection of Siderophores: CAS Agar Plate Technique

Siderophores production of the most efficient RP/TCP solubilizing isolates was determined using Chrome Azurol S (CAS) agar plate’s method described previously [34]. Agar plugs (10 mm diameter) of Actinobacterial cultures grown on solid SMM medium containing 0.5 g/L TCP as sole P source for 5 days at 28 °C were placed aseptically on CAS agar plate and incubated for 3 days at 30 °C. After incubation, the apparition of a yellow halo around the plugs indicated the production of siderophores [35].

### 2.7. Morphological, Physiological and Chemotaxonomic Characterization of the Selected Strains

The morphological, cultural, physiological and biochemical characteristics of the selected isolates were evaluated as described in the International Streptomyces Project [32]. Cultural characteristics were observed after growth on yeast extract–malt, extract agar (ISP2), oatmeal agar (ISP3) and inorganic salts–starch agar (ISP4) media for 7–21 days at 30 °C and the color series were determined as proposed previously [36]. The assimilation of carbohydrates was determined using the ISP9 medium containing 10 different carbohydrates, as sole carbon source, at a concentration of 1% (*w*/*v*). The chemical analyses of the diaminopimelic acid isomer were performed as described previously [37].

### 2.8. Amplification and Sequencing of the 16S rDNA of the Selected Strains

The 10 TCP/RP solubilizing Actinobacteria isolates were grown for 2 days at 28 °C in 500 mL flasks containing 100 mL of Hickey–Tresner medium (1 g/L yeast extract, 1 g/L beef extract, 2 g/L N-Z-Amine A (Sigma), 10 g/L Dextrin and 20 mg/L CoCl_2_∙6H_2_O) [38], under constant agitation of 180 rpm. Biomass was harvested by centrifugation (16,000× *g* for 10 min) and the mycelial pellet was used for automated extraction of DNA with the Maxwell^®^ RSC Instrument (Promega, Madison, WI, USA) and the Maxwell^®^ RSC PureFood GMO and Authentication Kit (Promega) according to manufacturer instructions.

The 16S rDNA was amplified with the PCR method using universal primers 27F (AGAGTTTGAMCCTGGCTCAG) and 1492R (GGTTACCTTGTTACGACTT). Amplification was carried out in 25 mL of reaction mixture containing 10 µL of AccuPower Taq PCR PreMix (Bioneer, Oakland, CA, USA), 1.25 µmol of each primer and 50 ng of DNA. PCR condition was as follows: after initial denaturation (96 °C for 1 min), 30 cycles of 96 °C for 30 s, 60 °C for 30 s and 72 °C for 1 min 30 s were performed, followed by a final extension (5 min, 72 °C). Amplification was carried out using a GeneAmp PCR 9700 System (Applied Biosystems, Foster City, CA, USA). Negative controls were included with no addition of template DNA. The amplified products were visualized on a 2% (*w*/*v*) agarose gel stained with ethidium bromide. PCR products from each isolate were sequenced using 27F and 1492R primers. Sequence similarity searches were performed against corresponding sequences of members of the Streptomycetaceae family using the online sequence analysis resources LEBIBI database [39] and GenBank through Nucleotide BLAST (http://www.ncbi.nlm.nih.gov/BLAST/ accessed: 29 March 2021). Unrooted phylogenetic tree was inferred using the Neighbor–Joining method [40]. The percentage of replicate tree in which the associated taxa clustered together in the bootstrap test (1000 replicates) is shown next to the branches [41]. The evolutionary distances were computed using the Kimura 2-parameter method [42] and are expressed in number of base substitutions per site. This analysis involved 52 nucleotide sequences. Evolutionary analyses were conducted in MEGA X [43].

### 2.9. Statistical Analysis

Statistical analysis of soil chemical parameters, total flora and Actinomycete strains distribution was carried out using ANOVA, and the Duncan test was used to compare the average abundance and percentage contribution of the Actinomycete isolates to the total flora in the three studied sites. All values are means of three replicates plates from the same soil sample. Least Significant Difference (LSD) was used to compare the parameter concentrations between sampling sites and standard deviation was calculated using SPSS software 20.0 package for Windows.

## 3. Results

### 3.1. Soil Analysis

Physico-chemical proprieties of soil samples are listed in Table 1. The tested sugar beet agricultural soils contained, on average, 5% of Organic Matter (OM) and 86% of Mineral Matter (MM) including 2% nitrogen, 3.5% P_2_O_5_, 0.35% K_2_O, MgO (1.4%) and MnO (0.08%). The pH of the soils was close to neutrality or slightly basic (pH 7.4 on average). The highest electrical conductivity indicating soil salinity was recorded in soil sample 2 with an average of 0.47 (µS/cm).

### 3.2. Distribution of Total Flora and of Actinomycetes in the Sugar Beet Fields under Study

The distribution of total flora (TF) and Actinomycetes of sugar beet soils collected from the three sites is shown in Table 2. TF was at a similar level in the three sites (on average 23 × 10^7^ cfu/g of soil) (Table 2). The Actinomycete isolates were significantly more abundant in sites 1 (7.2%) and 3 (6.73%) than in site 2 (4.90%) (Table 2).

Among the 164 Actinomycete isolates retained, 57 had the ability to grow on SMM + RP as the sole phosphate source. Among these 57 isolates, only 27 isolates (47.36%) had also the ability to grow on SMM + TCP as unique P source. Ten of the twenty-seven isolates, showing the highest biomass yield on SMM + RP and with different morphological characteristics, were selected for further studies. Seven of these ten Actinomycete isolates were from site 1 (AI, AYD, AV, AZ, BYC, BX, and BP), one (CYM) from site 2 and two from site 3 (DE1 and DE2).

### 3.3. Growth Kinetic of the Selected Actinomycete Strains in SMM + RP and SMM + TCP

Figure 2 clearly shows that the growth kinetics of most strains was similar in both TCP and RP. This indicated that the strains were able to assimilate similar amounts of phosphorus from these phosphate insoluble sources, with a comparable efficiency, and use it for their own growth. The only exception was the strain BP that showed a better growth on SMM + RP than on SMM + TCP. One notes that, in SMM+ TCP mainly, most strains yielded a lower biomass at day 5 than at day 4, suggesting cell lysis. However, the biomass yield was not the same for all strains and the strains could be grouped into two classes, class I with biomass yield above 70 µg/mL (DE2, BYC, AYD, AZ, AI and BP) and class II with biomass yield comprised between 50 and 70 µg/mL (AV, DE1, CYM and BX) (Figure 3). One notes that strains with the lowest biomass yields are also those yielding the lowest amount of soluble phosphate (Figure 3).

### 3.4. Estimation of the Amount Soluble Phosphate Released from TCP and RP by the Selected Actinomycete Strains

The concentration of free phosphate spontaneously released from TCP and RP in the control non-inoculated flask was 2.5 and 5 µg/mL, respectively. The concentration of soluble phosphate in the supernatant of most strains (except perhaps CYM and BX), exceeded this value (Figure 3). This indicated that most strains were able to release phosphate from these two different insoluble phosphate sources in excess of their phosphate need to support their growth. However, the amount of soluble phosphate released greatly varied with the nature of the phosphate source and from strain to strain.

Five strains released a high (>60 µg/mL) and higher amount of phosphate from TCP than from RP (Figure 3), indicating that TCP was more efficiently solubilized than RP. The presence of large amounts of phosphate in the supernatant of these strains in the presence of TCP simply indicated that the rate of Pi released from TCP exceeded its rate of consumption for bacterial growth.

Strain DE2 (class I) released maximal soluble phosphate concentration (180 µg/mL) from TCP, whereas strains BYC (class I) and BP (class I) released maximal amounts of soluble phosphate from RP (148.05 µg/mL and 59.44 µg/mL, respectively).

The strain BYC (class I) is of special interest since it was able to release a similar amount of Pi from RP (150 µg/mL) and TCP (170 µg/mL) and its biomass yield was similar with both phosphate sources (only12% higher in SMM + RP than in SMM + TCP). The strains BP (class I) and DE1 (class II) released over 2 fold more soluble phosphate from RP (60 µg/mL and 15 µg/mL, respectively) than from TCP (25 µg/mL and 7 µg/mL, respectively). The biomass yield of the strain BP was 20% higher in SMM + RP than in SMM + TCP whereas that of the strain DE1 was similar on both P sources. Interestingly, the strains BX (class II) and CYM (class II) consumed the totality of the phosphate released. This suggested that the solubilizing ability of these strains was less efficient than that of the others.

### 3.5. Cas-Agar Test and Evolution of the pH of the Growth Medium

The CAS-agar test indicated that the six more efficient TCP solubilizing Actinomycete strains (DE2, AYD, BYC, AZ, AI, BP) were producing siderophores (Figure 4) as previously reported for other PSM bacteria [20,44]. However, among these strains only BYC was able to efficiently release phosphate from RP. The strains BX, AV, DE1 and CYM apparently produced little siderophores (Figure 4) and were among the strains releasing very little phosphate from TCP or RP (Figure 3). This suggested that the production of siderophores contributes to the P solubilization process.

The pH of the growth medium of all strains in TCP as in RP felt between 6 and 6.5 at day 1 and raised afterwards in most cases. This indicated that the earliest solubilization process might involve the excretion of organic acids. The pH of the medium remained below 6 for the strains DE2, AZ and BX in TCP for the following days but reached 7 or above in RP. DE2 and AZ release a fair amount of P from TCP (180 µg/mL and 100 µg/mL, respectively) but a rather weak amount from RP (20 µg/mL) whereas BX did not release any P from any of these phosphate sources. DE2 and AZ possibly excreted more siderophores than BX (Figure 4).

In RP, the pH of the medium remained above 7 for most strains except for BYC (pH between 6.5 and 6.8 at days 2, 3 and 4) and CYM (pH 6 at days 1 and 2). BYC released a fair amount of phosphate from RP as well as from TCP (150 µg/mL and 170 µg/mL, respectively) whereas CYM hardly released any P from these two phosphate sources. BYC, but not CYM, was shown to excrete siderophores (Figure 4).

The pH of the medium of the 5 remaining strains (AYD, BP, AV, DE1 and AI) was rather similar in TCP and RP and above 7. AYD and AI, produced siderophores and released a fair amount of phosphate from TCP (140 µg/mL and 60 µg/mL, respectively) but a rather weak amount from RP (20 µg/mL). BP that is producing siderophores released more phosphate from RP (60 µg/mL) than from TCP (24 µg/mL). The latter two strains AV and DE1 were weak siderophore producers and yielded weak amount of phosphate from TCP (18 µg/mL and 7 µg/mL, respectively) as well as from RP (10 µg/mL and 14 µg/mL, respectively).

### 3.6. Taxonomical Characterization of the Selected Isolates

In order to determine whether the 10 isolates were similar or different strains, their ability to assimilate 10 different carbon sources was tested. Most strains were able to use mannitol, lactose, glucose, fructose, maltose, galactose, sucrose, sorbitol and glycerol as sole carbon sources, except AV that did not use mannitol, lactose, maltose, fructose and sorbitol; DE2, BYC, BP, AV and BX that did not use fructose; AYD that did not use maltose (Table 3) and DE2, BP and BX that did not use nor maltose nor fructose (Table 3). This preliminary analysis suggested that these strains were likely to be different.

These 10 isolates were also evaluated for their ability to withstand salt stress by growing at NaCl concentrations of 5, 7 and 10 g/L. All strains showed the best growth at 5 and 7 g/L NaCl (Table 3) except DE2, AYD and DE1, while CYM and BX showed better growth at 10 g/L NaCl (Table 3). Therefore, these strains could potentially be halotolerant. The analysis of cellular constituents of all isolates revealed the presence of L- diaminopimelic acid (DAP) isomer (Table 3), confirming that they belong to the *Streptomyces* genera.

The sequences of 16S rRNA gene of the 10 strains were analyzed using BLAST (http://www.ncbi.nlm.nih.gov/BLAST accessed: 29 March 2021) and the LEBIBI [39] and GenBank databases. They were all found to belong to the *Streptomyces* genus bearing an identity of at least 99%. Nucleotide sequences of partial 16S rRNA of the identified isolates were deposited into Gen-Bank Database (http://www.ncbi.nlm.nih.gov/GenBank/ accessed: 29 March 2021), under the accession numbers listed in Table 4.

16S rDNA sequences of *Streptomyces* species retrieved from Genbank as well as that of our strains were used for the construction of a phylogenic tree (Figure 5). Six strains (BP, BX, DE1, AV, AYD and AZ) were closely related to *Streptomyces bellus*, AI was related to *S. tunisiensis*, BYC to *S. enissocaesilis*, DE2 to *S. saprophyticus* and CYM to *S. cyaneofuscatus*.

## 4. Discussion

Most publications describing isolation of phosphate-solubilizing bacteria (PSB) used growth on TCP, rather than on RP, but analysis of scientific literature concerning biological P solubilization suggested that this substrate might not be the most appropriate to screen for PSB able to enhance plant growth [45,46]. In consequence, we instead used growth on RP as first screen to isolate PSM and our strategy and outcomes are summarized in Figure 6.

Our study revealed that 28% of the Actiomycetes isolated from sugar beet soils had the ability to grow on RP as sole phosphate source and among them only 47.36% had also the ability to grow on TCP as sole phosphate source. This difference is difficult to explain but suggested that the TCP and RP solubilizing processes may involve different mechanisms or that the greatest diversity of mineral elements present in RP compared to TCP, is necessary for the growth of some strains.

Five strains (DE2, BYC, AYD, AZ and AI) among the ten strains studied were able to release significant amount of phosphorus from TCP (>60 µg/mL). Their solubilization activity was comparable to that of *Azospirillum* sp., *Pantoea agglomerans* and *Pseudomonas fluorescens,* from wheat rhizosphere in Jensen medium [47]; however, it was much higher than that reported for a *Streptomyces* sp. (AH6 strain) isolated from the rhizospheric soil of *Calluna vulgaris* L [20,48].

However, among these five efficient solubilizing strains, only one strain, BYC related to *Streptomyces enissocaesilis* (Figure 5), was able to release similar amount of phosphorus from RP and TCP and another strain, BP, was able to release even more phosphate from RP than from TCP. These two strains are thus of special interest and the mass spec analysis of the molecules present in their supernatant is expected to lead to the purification and structural characterization of potentially novel siderophores and/or organic acids contributing to their efficient RP solubilization process.

In RP the pH of the medium of all strains, except that of BYC, was between 7 and 8, suggesting that the solubilization process does not involve the excretion of organic acids but rather that of siderophores [41,49]. The very efficient solubilization process of BYC might involve both the acidification of the growth medium and the excretion of siderophores.

In TCP, the growth medium of DE2 and AZ turned out to be acidic but that of the other strains was close to or above 7. This suggested that the TCP solubilization process of DE2 and AZ might involve the excretion of organic acids. To our knowledge, that is the first report of Actinomycete strains solubilizing insoluble P via the production of organic acids. In this process, negatively charged organic acids are thought to chelate Ca^2+^ counter ions of phosphate [43,47]. A similar process was reported in other fungi such *Penicillium aurantiogriseum* [50] and *Penicillium radicum* [41,47,49,51].

In conclusion, we anticipate that our most efficient RP solubilizing strain, BYC, produced in large scale may constitute a novel kind of fertilizers useful to solubilize the phosphate trapped in the soil to feed and stimulate sugar beet growth. Such strategy would contribute to the development of a bio-based economy supporting a sustainable and environmentally friendly agriculture.

## Figures and Tables

**Figure 1 microorganisms-09-00914-f001:**
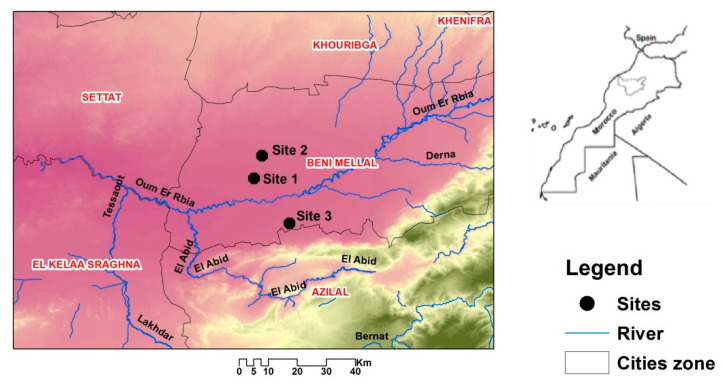
Map of the sampling sites of the three sugar beet agricultural soils of the Beni-Mellal region of Morocco.

**Figure 2 microorganisms-09-00914-f002:**
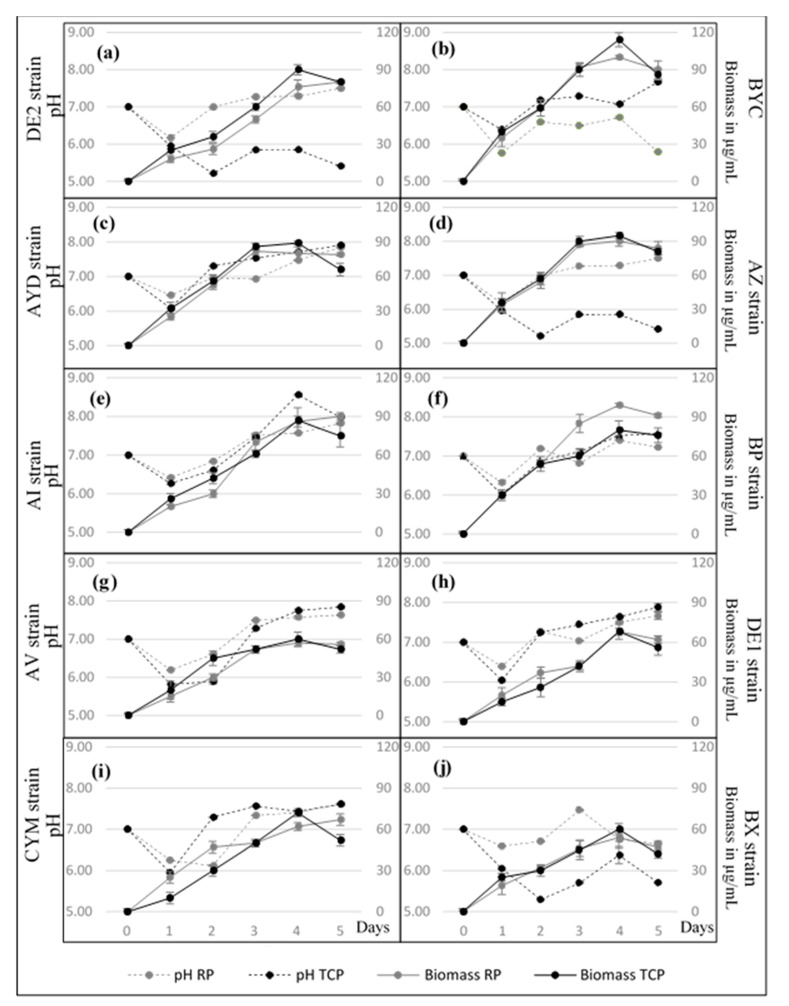
Evolution of the biomass (continuous lines) and the pH of the medium (dotted lines) of the selected Actinomycete isolates grown in SMM+ TCP (black circle) and SMM+ RP (gray circle). Error bars represent standard deviations of the mean values of the results of three independent culture replicates.

**Figure 3 microorganisms-09-00914-f003:**
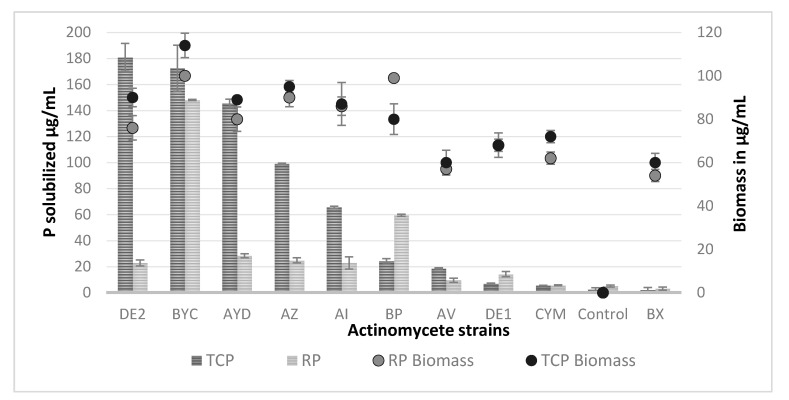
Concentration of soluble phosphate released (µg/mL) from TriCalcium Phosphate (TCP, dark gray histograms) and natural Rock Phosphate (RP, light gray histograms) in the non-inoculated flasks (control) and in the supernatant of cultures of the ten selected Actinomycete isolates grown for five days in SMM containing 0.5 g/L RP or 0.5 g/L TCP. The maximal biomass yield in µg/mL of the 10 selected Actinomycete strains is shown above the histograms for TCP (black circles) and RP (gray circles). Error bars represent standard deviations of the mean values of the results of three independent culture replicates.

**Figure 4 microorganisms-09-00914-f004:**
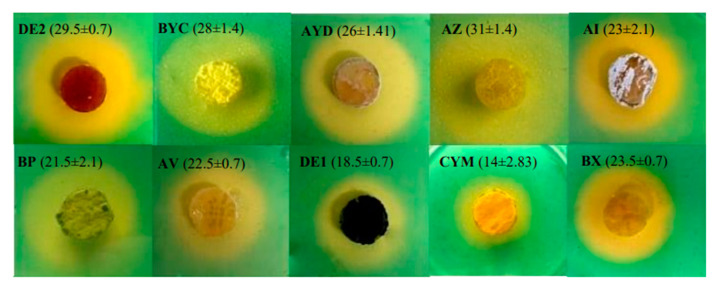
Halo of discoloration of CAS-agar around agar plugs of the ten most efficient TCP/RP solubilizing Actinomycete isolates originating from sugar beet soils of the Beni-Mellal region grown for five days in solid SMM containing 0.5 g/L TCP and deposited on the surface of a CAS–blue agar plate. The formation of the halo around the plugs is thought to be due to the excretion of siderophores. Diameters of the halos expressed in mm are shown in proximity of the name of the isolates.

**Figure 5 microorganisms-09-00914-f005:**
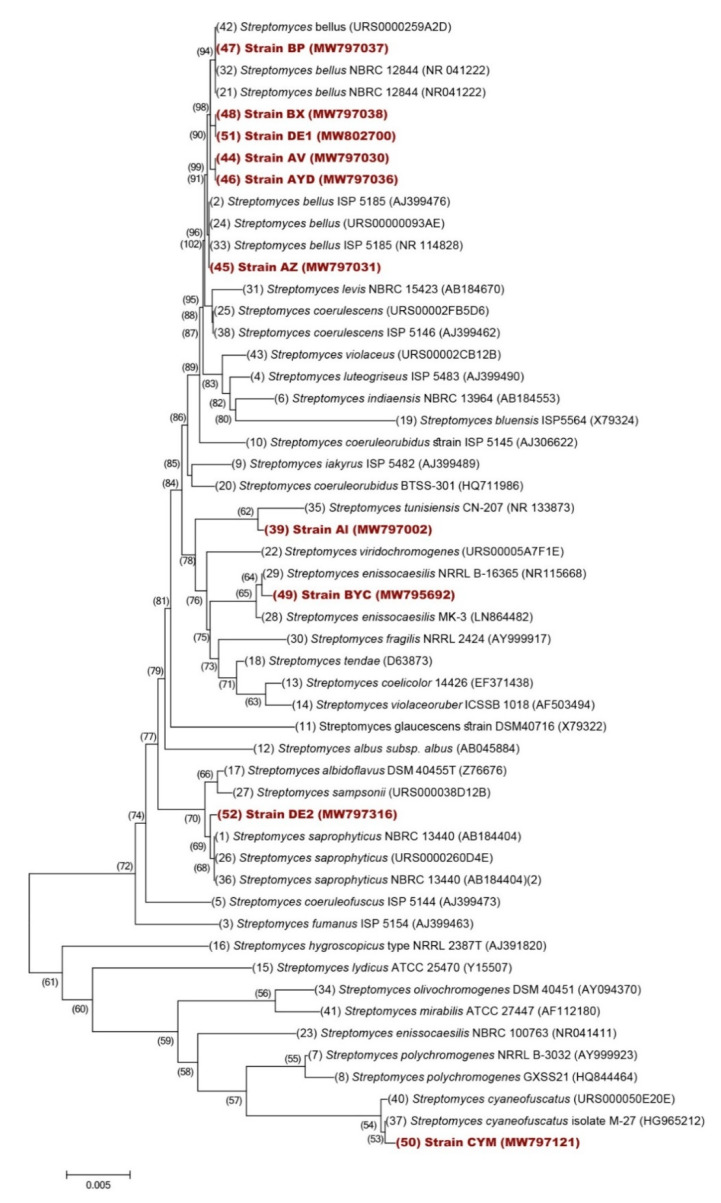
Neighbor–Joining phylogenetic tree of the 10 isolated strains and 42 *Streptomyces* species based on nearly complete 16S rRNA gene sequences (1400 nt). Numbers at nodes indicate levels of bootstrap support (%) based on a Neighbor–Joining analysis of 1000 resampled datasets; only values >50% are given. Accession numbers are given in parentheses. Bar, 0.005 nucleotide substitutions per site.

**Figure 6 microorganisms-09-00914-f006:**
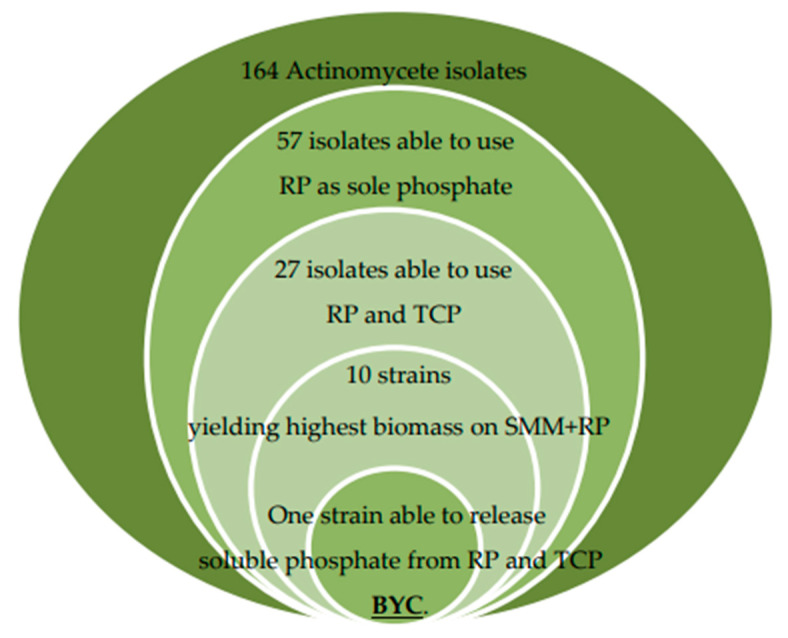
Schematic representation of the experimental strategy used to isolate efficient TCP/RP solubilizing Actinomycete strains.

**Table 1 microorganisms-09-00914-t001:** Physico-chemical characteristics of the three sugar beet soils of the Beni-Mellal region.

Parameters	Site 1	Site 2	Site 3	ANOVA
Perimeter	Beni Amir		Beni Moussa	
Graphical situation	3240′10.8” N, 6°84′14” W	3247′34.45” N, 6°81′52.3” W	3226′31.29” N, 6°73′36.30” W	
pH_water_	7.44 ± 0.27 ^a^	7.31 ± 0.26 ^a^	7.49 ± 0.14 ^a^	
pH_KCL_	6.49 ± 0.16 ^a^	6.93 ± 0.06 ^b^	7.09 ± 0.04 ^c^	*p* < 0.001
Organic matter (%)	4 ± 0.3 ^a^	6 ± 0.3 ^b^	4 ± 0.2 ^a^	*p* < 0.001
Mineral matter (%)	89 ± 0.2 ^a^	81 ± 0.2 ^b^	88 ± 0.1^c^	*p* < 0.001, *p* = 0.026
Water content (%)	7.4 ± 0.2 ^a^	1.2 ± 0.1 ^b^	7.2 ± 0.1 ^a^	*p* < 0.001
Electrical conductivity (µS/cm)	0.40 ± 0.13 ^a^	0.47 ± 0.009 ^ab^	0.25 ± 0.002 ^ac^	*p* = 0.015
Total nitrogen (%)	2.62 ± 0.00 ^a^	1.75 ± 0.87 ^a^	2.33 ± 0.50 ^a^	
P_2_O_5_ (%)	4.96 ± 1.22 ^a^	5.23 ± 1.24 ^a^	0.34 ± 0.02 ^b^	*p* = 0.001
Magnesium oxide MgO (%)	0.95 ± 0.16 ^a^	1.00 ± 0.11 ^a^	2.22 ± 0.28 ^b^	*p* < 0.001
Potassium oxide K_2_O (%)	0.25 ± 0.03 ^a^	0.04 ± 0.03 ^b^	0.77 ± 0.14^c^	*p* = 0.026, *p* < 0.001
Manganese oxide MnO (%)	0.10 ± 0.03 ^a^	0.09 ± 0.01 ^a^	0.04 ± 0.006 ^b^	*p* = 0.009, *p* = 0.019

Different letters indicate significant differences at *p* < 0.05. LSD test was used to compare the parameter concentrations between sampling sites.

**Table 2 microorganisms-09-00914-t002:** Distribution of total flora and of Actinomycete strains in the three sugar beet sites of the Beni-Mellal region.

	Site 1	Site 2	Site 3	ANOVA
Perimeter	Beni Amir		Beni Moussa	
Total flora (×10^7^ UFC/g of soil)	25.66 ^a^	25.91 ^a^	17.08 ^b^	
Actinomycetes (×10^7^ UFC/g of soil)	1.84 ^a^	1.27 ^ab^	1.15 ^b^	*p* = 0.03
% des Actinomycetes/Total flora	7.2%	4.90%	6.73%	

Different letters indicate significant differences at *p* < 0.05. Duncan t-test was used to compare mean percentages and Actinomycetes density.

**Table 3 microorganisms-09-00914-t003:** Biochemical and morphological characteristics of the 10 selected isolates.

TCP Solubilizing Actinomycete Isolates 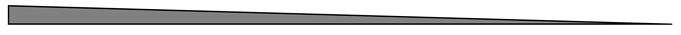
Origin	Site 3	Site 1	Site 3	Site 2	Site 1
Strains	DE2	BYC	AYD	AZ	AI	BP	AV	DE1	CYM	BX
ISP 3	+++	++	++	++	++	++	++	+	+	+
ISP4	+++	–	–	–	–	–	–	–	–	–
ISP6	+	–	–	–	+	–	–	–	–	+
Aerial spore mass	White	White	Gray	Gray	Green	White	White	Clear green	White	White
Soluble pigment	Yellow	–	–	–	–	–	–	Yellow	–	–
Colony reverse	Yellow	Bright yellow	Brown	Yellow	Clear Gray	Yellow	Brown	Brown	Yellow	Green
DAP-isomer	LL	LL	LL	LL	LL	LL	LL	LL	LL	LL
C. source utilization	
Mannitol	+	+	+	+	+	+	–	+	+	+
Lactose	+	+	+	+	+	+	–	+	+	+
Glucose	+	+	+	+	+	+	+	+	+	+
Maltose	–	+	–	+	+	–	–	+	+	–
Galactose	+	+	+	+	+	+	+	+	+	+
Glycerol	+	+	+	+	+	+	+	+	+	+
Sucrose	+	+	+	+	+	+	+	+	+	+
Citrate	+	+	+	+	+	+	–	–	+	–
Fructose	–	–	+	+	+	–	–	+	+	–
Sorbitol	+	+	+	+	+	+	–	+	+	+
Growth at different concentration of NaCl (g/L)	
5(g/L)	++	+++	+	+++	++	+++	++	+++	+++	+++
7(g/L)	+	++	+++	+++	+++	+++	+++	+++	+++	+++
10(g/L)	++	++	+	++	++	+	–	+++	+++	+++

+: Tested positive/utilized as substrate; –: tested negative/not utilized as substrate; +++: Strong growth/production; ++: Medium growth/production; +: Low growth/production.

**Table 4 microorganisms-09-00914-t004:** 16S rRNA identification of the 10 selected isolates.

Strains	16S rRNA Identification	Accession Number
AYD	*Streptomyces bellus*	MW797036
DE2	*Streptomyces saprophyticus*	MW797316
BYC	*Streptomyces enissocaesilis*	MW795692
AI	*Streptomyces tunisiensis*	MW797002
AZ	*Streptomyces bellus*	MW797031
BP	*Streptomyces coerulescens*	MW797037
AV	*Streptomyces bellus*	MW797030
CYM	*Streptomyces cyaneofuscatus*	MW797121
BX	*Streptomyces bellus*	MW797038
DE1	*Streptomyces bellus*	MW802700

## Data Availability

The data that support the findings of this study are available from the corresponding author upon reasonable request.

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
