# Peer review of "Isolation and Characterization of Phosphate Solubilizing Streptomyces sp. Endemic from Sugar Beet Fields of the Beni-Mellal Region in Morocco"

_microorganisms, 2021, doi:10.3390/microorganisms9050914_

Round 1

Reviewer 1 Report

In this paper, Aallam and coworkers describe the isolation and characterization of phosphate solubilizing Streptomyces from sugar beet fields in Morocco.

The work is well written in each of its sections. I only suggest reporting a brief description of the aim of the research in the abstract in order to address the reader to the core of it. In the M&M section, report that you performed the statistical analysis also for soil analysis.

Check carefully grammar and syntax.

Author Response

Response to Reviewer 1 Comments

Point 1 : I only suggest reporting a brief description of the aim of the research in the abstract in order to address the reader to the core of it.

Response 1 : Correction done in the text

Point 2 : In the M&M section, report that you performed the statistical analysis also for soil analysis.

Response 2 : Yes, we specified it in the M&M section.

Point 3 : Check carefully grammar and syntax.

Response 3 : Done

Reviewer 2 Report

This original research article submited  to Microorganisme (MDPI) by Dr. Allam and colleagues describe the isolation and characterisation of phosphate solubilising Streptomyces isolated in Morocco.

The author selected three isolation sites from which they analysed the physico-chemical characteristics of the soil. Then they carried out isolation proceidures of actinomycetes with classic microbiological approach and screened 164 isolated actinomycetes for their ability to use Rock Phosphate. Such strains were tested for their ability to grow on TCP and the 10 best candidates were shortlisted on the basis of the morphologic diversity. A thourough kinetic analysis of P solubilisation in liquid culture was performed with these candidate, enabling to identify the 6 most active strains. These 6 strains also released siderophores according to CAS assay. The 10 Streptomyces were phenotypically (ISP media, C, source, colour) and taxonomically  (16S based) characterised.

Overall the manuscript is clear and well written, the experiments have been conducted in a technically sound manner and the figue layout is very professional. I would suggest that the author address the points below to improve the manuscript.

Major Points

L37 : intrans is not an english word, please prefer fertiliser ?

L110 : Does this reference is accurate ?

L115 : is it mg or ml ?

L213 : the total flora of the soil cannot be assessed on a actinomycete selecting media, maybe replace by the « total number of isolate compatible with growth on AIA media » or similar phrasing

L370 : this claim should  be downtuned as the author did not prove by any mechanistic insights how their strains solubilise P either by organic acid or siderophores. In addition, other publication already raised their hypothesis, see Franco Correa et al., Applied soil ecology 45(3) 2010.

Minor Points

L97 : the soil was treated, please rephrase by the soil was used for further experiments within 48 hours

L122 : what is the origin of the TCP used

L129 : there is a typo

L159 : write N-Z-Amine according to Sigma reference

L200 : what is the number of replication in soil analysis

Author Response

Response to Reviewer 2 Comments

Point 1 : L37 : intrans is not an english word, please prefer fertiliser ?

Response 1 : Correction done in the text

Point 2 : L110 : Does this reference is accurate ?

Response 2 : Yes the reference [29] is accurate

Point 3 : L115 : is it mg or ml ?

Response 3 : It is ml ; Two grams (wet weight) of each soil sample were suspended in 18 ml of sterile physiological serum (9 g/l, NaCl).

Point 4 : L213 : the total flora of the soil cannot be assessed on a actinomycete selecting media, maybe replace by the « total number of isolate compatible with growth on AIA media » or similar phrasing

Response 4 : Yes, we specified it in the M&M section.

We plated our soil suspensions on nutrient agar plates (Difco, USA) known to allow growth of Gram positive and Gram negative bacteria and on synthetic minimum medium (SMM) specifically used for Actinomycetes isolation as described previously (Hamdali et al., 2008).

Point 5 : L370 : this claim should  be down tuned as the author did not prove by any mechanistic insights how their strains solubilise P either by organic acid or siderophores. In addition, other publication already raised their hypothesis, see Franco Correa et al., Applied soil ecology 45(3) 2010.

Response 5 : The reviewer is right to consider that we did not provide a direct proof that the siderophores or the acids produced did cause insoluble phosphate solubilisation since we did not purified these substances to provide a rigorous proof. However, we consider that is likely since many reports in the literature mention that organic acids or siderophores are able to destabilize the structure of rock phosphate thanks to their abilities to chelate the counter ions of phosphate. However we tuned down our claim as requested by simply saying “The very efficient solubilization process of BYC might involve both the acidification of the growth medium and the excretion of siderophores.”

Minor Points

Point 6 : L97 : the soil was treated, please rephrase by the soil was used for further experiments within 48 hours.

Response 6 : Correction done in the text.

Point 7 : L122 : what is the origin of the TCP used

Response 7 : Yes, we specified it in the text (Page 4, L139):

TCP (0.5 g/l, Ca3 (PO4 )2) (Sigma Aldrich, France)

Point 8 : L129 : there is a typo

Response 8 : Correction done

Point 9 : L159 : write N-Z-Amine according to Sigma reference

Response 9 : Correction done

Point 10 : L200 : what is the number of replication in soil analysis

Response 10 : Three replicates as indicated in M&M.